# Memories in a network with excitatory and inhibitory plasticity are encoded in the spiking irregularity

**Júlia V. Gallinaro** *, **Claudia Clopath**

Bioengineering Department, Imperial College London, London, United Kingdom

* j.gallinaro@imperial.ac.uk

## Abstract

Cell assemblies are thought to be the substrate of memory in the brain. Theoretical studies have previously shown that assemblies can be formed in networks with multiple types of plasticity. But how exactly they are formed and how they encode information is yet to be fully understood. One possibility is that memories are stored in silent assemblies. Here we used a computational model to study the formation of silent assemblies in a network of spiking neurons with excitatory and inhibitory plasticity. We found that even though the formed assemblies were silent in terms of mean firing rate, they had an increased coefficient of variation of inter-spike intervals. We also found that this spiking irregularity could be read out with support of short-term plasticity, and that it could contribute to the longevity of memories.

## Author summary

Memories are thought to be stored in the brain in groups of strongly connected neurons. In some cases, these memories can exist in a dormant state and be active exclusively when activated by an external stimulus. In this article, we show that even when they exist in a dormant state, there could be a trace left in how neurons are firing. This trace is left not in terms of total activity level, which is why we consider the memory to be silent, but in terms of firing pattern, or more specifically how irregular the individual spike trains are. This suggests that these memories could be used for further processing even when they exist in a dormant state, without the need of being reactivated by an external stimulus. Furthermore, we also show how this irregular firing pattern contributes to the longevity of the memory by allowing a slower decay of weights between strongly connected neurons.

## Introduction

Cortical synapses are plastic, allowing sensory experience to be stored in network connectivity. Concurrent activation of ensembles of neurons is thought to promote cell assembly formation by potentiating their synapses [1]. Stronger synapses within neurons allows them to be more easily activated together, even in the presence of partial cues. How exactly cell assemblies are

**Data Availability Statement:** All code is available on GitHub (https://github.com/juliavg/EI_assembly).

**Funding:** This work was supported by BBSRC BB/N013956/1, BB/N019008/1, Wellcome Trust

200790/Z/16/Z, Simons Foundation 564408 and
EPSRC EP/R035806/1 (all to CC). The funders had
no role in study design, data collection and
analysis, decision to publish, or preparation of the
manuscript.

formed and how their synapses and firing activity encode information, however, is yet to be fully understood. Theoretical work [2–8] has shown it is possible to create such assemblies by combining different forms of synaptic plasticity. In some of these models [3, 4], when strongly connected assemblies are formed, spontaneous activity is characterized by an overall stable firing rate across the excitatory population, but with the firing rate of individual assemblies transitioning between periods of high and low activity.

Cell assemblies, however, may not necessarily be persistently active at all times. Assemblies could also be stored in a latent or quiescent state, from where they can be retrieved by a cue [9]. Storing them in a silent state could be advantageous from an energy efficiency point of view [10], specially if they are not being constantly recalled. Inhibitory engrams have been proposed as a way of implementing this type of silent assembly, and were suggested to form when increased excitation within a highly active ensemble of neurons would be matched by increased inhibition [11]. Theoretical work [2, 5, 6] has shown such silent assemblies can be formed by combining traditional spike-timing dependent forms of excitatory plasticity [12–14] with inhibitory plasticity [2, 15, 16]. In these models, inhibitory plasticity counteracts the effect of excitatory potentiation, leading to the formation of cell assemblies in which the excitatory neurons receive increased excitatory and inhibitory currents (EI assemblies).

Although silent EI assemblies do not reactivate themselves during spontaneous activity, the memories they encode can still be reactivated by specific stimulation. Vogels, Sprekeler et al. [2] have shown that memories could be retrieved by momentarily disrupting balance within the assembly by stimulating a fraction of their neurons. Similarly, in Yger et al. [5], strengthening of the assembly led to stronger neural response upon stimulation. Since neurons belonging to an EI assembly receive increased excitation and inhibition, memories encoded by the EI assembly could also be reactivated by disinhibition [17, 18]. A transient decrease in inhibitory drive leads to a net increase in excitatory input to excitatory neurons belonging to the assembly, resulting in an increase in their activity. In Barron et al. [17], for example, dormant memories embedded in a network model could be retrieved by decreasing the efficacy of inhibitory synapses.

Here, we study how stronger connectivity within EI assemblies influences spontaneous activity. More specifically, we show that neural stimulation does form EI assemblies that are silent in terms of firing rate, but leaves a trace in the regularity of their spike trains. Therefore, we suggest it is possible to read out assembly information not only with specific stimulation, but also during spontaneous activity. In a feedforward model, we show that an increase in excitatory current leads to an increase in irregular firing in a neuron receiving feedforward plastic inhibition. We also show this irregularity can be read out with support of short-term plasticity (STP) [19]. We extend these results to a network model with excitatory [13] and inhibitory [2] plasticity, and show that neurons belonging to an EI assembly indeed fire more irregularly. During spontaneous activity, we demonstrate that irregularity can be read out with STP, even though assembly neurons are not being specifically stimulated and their mean firing rate is indistinguishable from the other excitatory neurons in the network. Furthermore, we analyze the decay of excitatory weights, and find that memory lifetime is increased due to the irregular firing of the neurons within the EI assemblies. Put together, our results suggest that, in silent assemblies, memories may be encoded in regularity of firing during spontaneous activity, which allows them to be read out without specific stimulation, and that this could contribute to their longevity.

## Results

### iSTDP leads to more irregular firing upon increased excitatory currents

Inhibitory plasticity has been previously proposed as a mechanism to promote balance between excitatory and inhibitory currents to a neuron [2, 20], promoting homeostasis of the

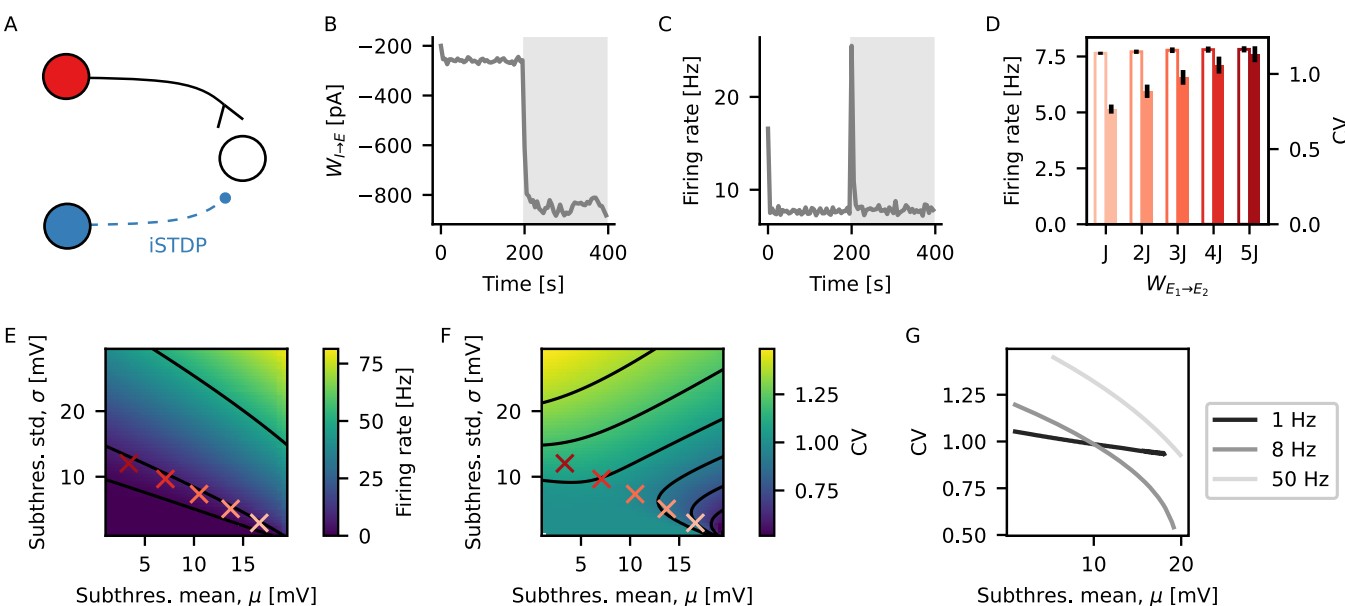

**Fig 1. iSTDP leads to more irregular firing upon increased excitatory currents.** (A) A single LIF neuron receives input from one excitatory source with a fixed weight, and one inhibitory source through iSTDP. (B) Synaptic weight from the inhibitory source to the output neuron as a function of time. The grey shaded area indicates the period where the weight from the excitatory source is increased from $W_{E_1 \to E_2} = J$ to $W_{E_1 \to E_2} = 4J$. (C) Firing rate of the output neuron as a function of time. Grey shaded area as in (B). (D) Mean firing rate (edge colored) and CV (full colored) of the output neuron for different values of increase in excitatory current $W_{E_1 \to E_2} = J, 2J, 3J, 4J, 5J$. Black lines show standard deviation across 10 independent output neurons. Firing rate and CV are calculated using the last 50 s of the simulation. (E) Predicted firing rate of an LIF neuron as a function of mean ($\mu$) and standard deviation ($\sigma$) of its subthreshold membrane potential from theoretical calculations [23]. The red crosses indicate mean and standard deviation of subthreshold membrane potential from neurons in (D), with matching colors. Black lines show contour lines for firing rate equal 1, 8 and 50 Hz. (F) Same as (E) for CV. (G) Predicted CV from theory as a function of mean subthreshold membrane potential for a fixed firing rate, matching the contour lines on (E).

post-synaptic firing rate [2]. Such homeostatic regulation of firing rate at different levels of input currents, however, could have an effect on higher order statistics of post-synaptic firing. We therefore started by testing the effect of inhibitory plasticity on the irregularity of firing when a neuron received different intensities of excitatory current. For that, we used the inhibitory spike-timing dependent plasticity model (iSTDP) proposed by Vogels, Sprekeler et al. [2], which has been previously shown to reproduce multiple experimental results [2, 21, 22]. We simulated a single LIF neuron that received input from one excitatory source with a fixed weight $W_{E_1 \to E_2} = J$, and from one inhibitory source through iSTDP [2] (Fig 1A). After the inhibitory weight had reached an equilibrium value, and the neuron fired at target rate, we increased the strength of the excitatory connection $W_{E_1 \to E_2}$. As expected, just after an increase of the excitatory current, the output neuron fired at a higher rate, triggering an upregulation of the inhibitory weight by iSTDP (Fig 1B and 1C), until the output neuron fired again at target rate (Fig 1C). We then repeated this procedure systematically, for different values of increase in strength of the excitatory connection $W_{E_1 \to E_2} = 2J, 3J, 4J, 5J$. We observed that after plasticity had converged, the neuron always fired at the target rate, but with a coefficient of variation of inter-spike intervals (CV) that increased with $W_{E_1 \to E_2}$ (Fig 1D).

To better understand this result, we calculated the expected firing rate and CV for an LIF neuron as a function of the mean and variance of its subthreshold membrane potential [23, 24] (see Methods for details of the calculation). Although higher excitatory and higher inhibitory currents contribute to the mean with different signs, they both contribute positively to the variance [23, 24]. This means that the same firing rate can be achieved by different combinations of mean and variance of the subthreshold membrane potential (contour lines on Fig 1E).

At the same time, different combinations of mean and variance of the subthreshold membrane potential will lead to different values of CV (Fig 1F). More specifically, for a given fixed firing rate, the CV will be higher when mean is lower and variance is higher (Fig 1G). Therefore, for a neuron receiving inhibitory plastic input under iSTDP, an increase in excitatory currents leads to lower mean membrane potential and more irregular spikes.

## Different levels of irregularity can be read out with short-term plasticity (STP)

If the irregularity of spike trains can carry information about previous stimulation, one important question is whether it can be decoded by an output neuron. To that end, we connected an output LIF neuron to multiple inputs with same rate and CV (Fig 2A). For a given input firing rate, the postsynaptic subthreshold membrane potential had a constant mean $\mu_{V_m}$ (Fig 2B), and a standard deviation $\sigma_{V_m}$ that increased with CV (Fig 2C). The increase in $\sigma_{V_m}$ alone, however, was not enough to trigger large modulation of output firing rate with input CV. Therefore, we found that an increase in CV of the input neurons led to slightly increased firing rate of the output neuron (Fig 2D).

Previously, STP has been shown to increase postsynaptic sensitivity to bursts [25]. Here, introducing short-term facilitation [19] (STF) in the connections to the output neuron (Fig 2E) led to modulation of the mean $\mu_{V_m}$ (Fig 2F), as well as the standard deviation $\sigma_{V_m}$ (Fig 2G), of the subthreshold voltage with the CV of the input neurons. This in turn was reflected in a larger modulation of output rate with input CV (Fig 2H).

The weight of the static synapses (Fig 2A–2D) was chosen to be equal to the mean plastic weight from the simulation with STF (Fig 2E–2H) when CV = 0.4 (see S1(A) Fig for a distribution of the plastic weights). Increasing the static weight to match that of CV = 1.4 led to similar results (S1(B)–S1(D) Fig).

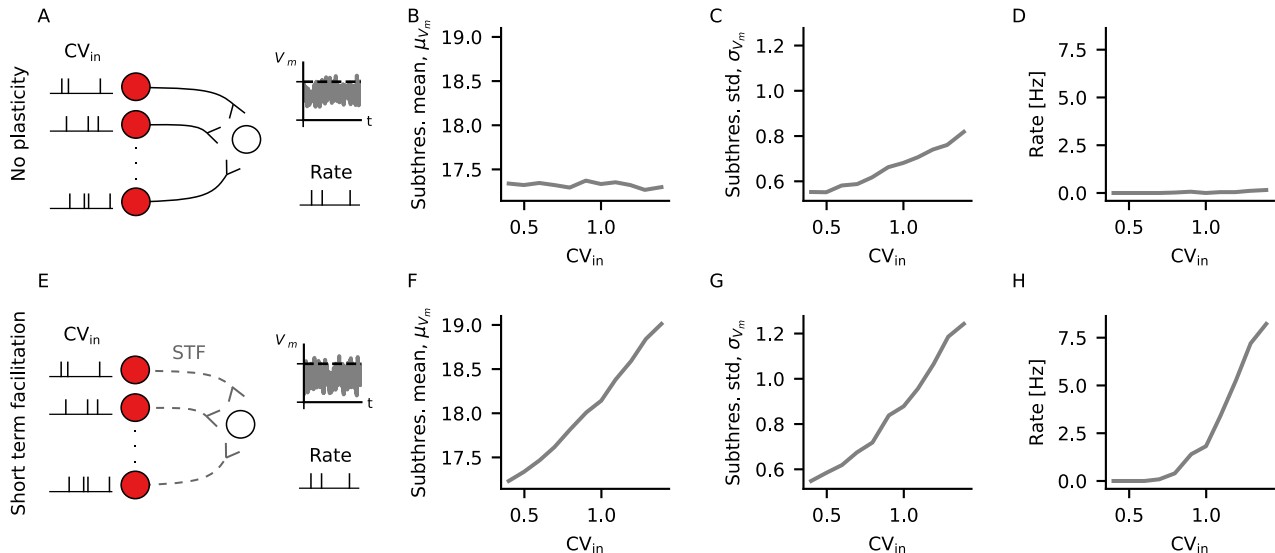

**Fig 2. Different levels of irregularity can be read out with short-term plasticity (STP).** (A) An output neuron receives input through static excitatory connections from multiple neurons firing at the same rate and CV. The small plots on the right illustrate what is measured from the output neuron, namely its subthreshold membrane potential and its firing rate (B) Mean subthreshold membrane potential of the output neuron in (A), for different values of CV. (C) Standard deviation of the subthreshold membrane potential of the output neuron in (A) for different values of CV. (D) Output firing rate of the neuron in (A) for different values of CV. (E-H) Same as (A-D), but with output neuron receiving input through plastic excitatory connections following short-term facilitation.

In summary, although higher values of input CV lead to larger standard deviation of the subthreshold voltage of the output neuron in the presence of static synapses, this has only a small effect on the firing rate of the output neuron. In the presence of STF, on the other hand, higher input CV also leads to higher mean subthreshold voltage of the output neuron, leading to a larger modulation of the output rate with input CV (see also S2 Fig). Therefore, the irregularity of spike trains can be decoded with the support of STF.

## Assemblies formed by excitatory and inhibitory plasticity are silent but leave a trace in terms of irregular firing

Inhibitory plasticity has been proposed to support the formation of balanced excitatory-inhibitory assemblies (EI assemblies) by matching high excitatory currents in neurons following increased excitatory plasticity [2, 11]. We just showed that an increase in excitatory current led to more irregular firing of a neuron receiving inhibitory input through iSTDP (Fig 1) and that a difference in irregularity modulated the output firing rate of a neuron receiving plastic input under STF (Fig 2). Put together, this suggests that formation of EI assemblies leaves a trace on the regularity of spike trains, which can be read out with STP.

In order to test this idea, we started by investigating whether the formation of EI assemblies left a trace on irregularity of firing in a model similar to the one presented in Vogels et al. [2]. We simulated a recurrent network of excitatory and inhibitory LIF neurons in which inhibitory-to-excitatory synapses were plastic according to the iSTDP rule [2] and other synapses were static (S3(A) Fig). We also included two output neurons, one receiving input from the neurons within the EI assembly and the other receiving input from a group of excitatory neurons outside the assembly (S3(A) Fig). Both output neurons received those inputs through connections that were plastic according to STF. Once the excitatory neurons had reached their target firing rate, we formed an assembly by hardwiring an increase in excitatory weights between assembly neurons by a factor of 6. As shown in Vogels et al. [2], this increase in recurrent excitatory weights led to an increase in firing rate of assembly neurons, which triggered an increase in incoming inhibitory weights through the iSTDP rule [2] (S3(B) Fig). Following a transient period, the within-assembly excitatory neurons fired again at target rate (S3(C) and S3(D) Fig), but with higher CV (S3(E) and S3(F) Fig). Moreover, following the assembly formation, the output neuron connected to the assembly fired with higher rate than the one connected to excitatory neurons outside the assembly (S3(H) Fig).

We proceeded by including excitatory plasticity on this recurrent network, such that assemblies could be formed by specific stimulation of neurons. Starting from the previous model of recurrent network (S3(A) Fig), we made excitatory-to-excitatory synapses plastic according to a triplet-based model of STDP [13] (Fig 3A). This triplet model was built as an extension of classical pair-based STDP models, and has been shown to reproduce a series of experiments on plasticity [13]. In this model, weights are potentiated by post-pre-post triplets, such that high post-synaptic firing rate leads to LTP [13]. An EI assembly was then formed by stimulating a subset of the excitatory neurons (Fig 3A). The increase in activity following stimulation led to potentiation of both excitatory synapses through the triplet rule and inhibitory synapses through iSTDP, as shown in Vogels et al. [2] (Fig 3B, S4(A) and S4(B) Fig). After a transient period, given the homeostatic nature of the iSTDP rule [2], the mean firing rate of the stimulated neurons was indistinguishable from the rest of the network (Fig 3C and 3D). At this point, due to the increase in synaptic weights (Fig 3B), the neurons belonging to the assembly received more excitatory and more inhibitory currents than before the stimulation protocol, which led to more irregular spike trains (Fig 3E and 3F and S4(D) Fig).

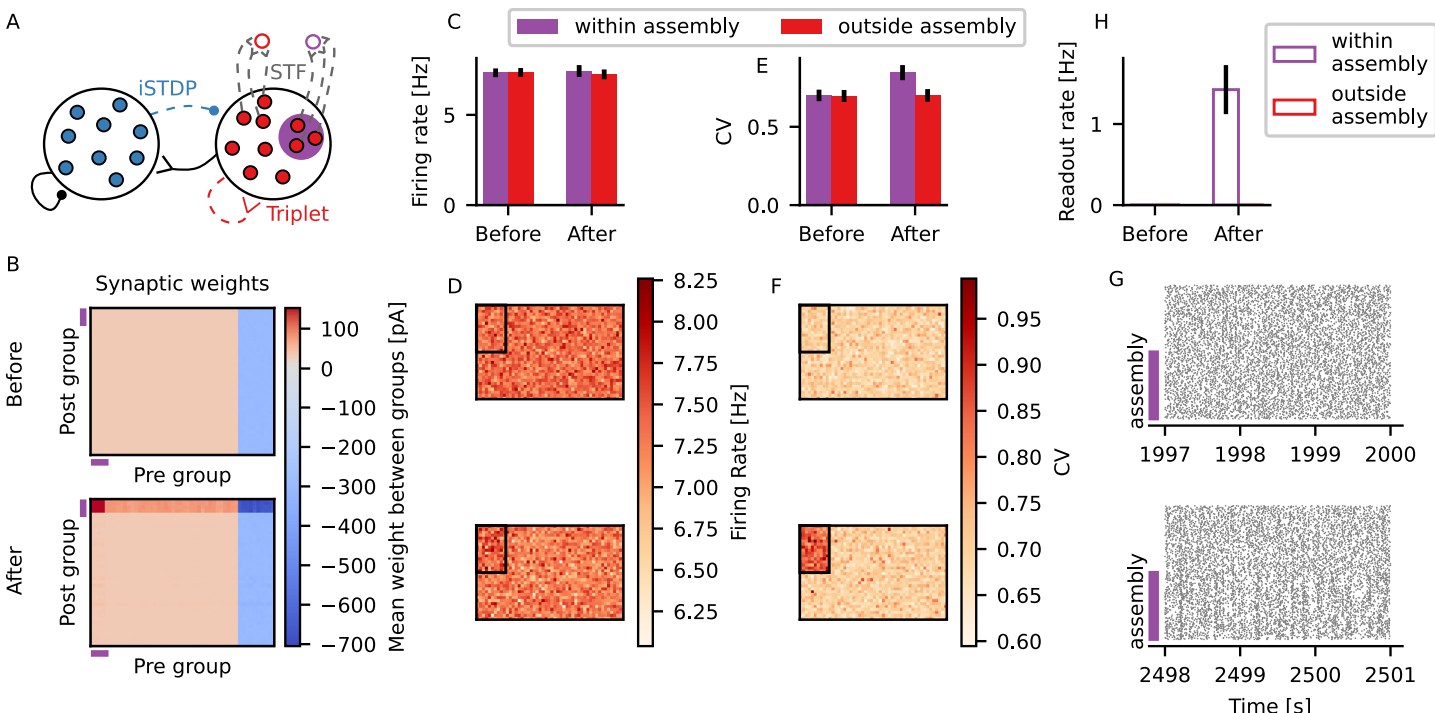

**Fig 3. Assemblies formed by excitatory and inhibitory plasticity are silent but leave a trace in terms of irregular firing.** (A) The simulated network is composed of excitatory (red) and inhibitory (blue) LIF neurons. Inhibitory-to-excitatory connections (dotted blue) are plastic according to iSTDP [2] and excitatory-to-excitatory (dotted red) according to the triplet rule [13]. An EI assembly is formed when a subset of the excitatory neurons (purple) is stimulated. One readout neuron receives input from the EI assembly (purple edge color) and another readout neuron receives input from a subset of excitatory neurons outside of the assembly, but with same size as the assembly (red edge color). The readout synapses are plastic under STF (dotted grey). Inhibitory-to-inhibitory and excitatory-to-inhibitory connections (black) are static. (B) Mean synaptic weight between groups of neurons. Neurons are sorted such that the first 160 neurons are assembly neurons. Neurons are then divided into groups of 40 neurons, and shown is the average synaptic weight between groups. Shown are excitatory (red scale) and inhibitory (blue scale) synaptic weights to excitatory neurons only. Synaptic weights which are not plastic are not shown. The purple lines indicate the position of groups comprising assembly neurons only. (C) Mean firing rate of neurons within the assembly (purple) and outside the assembly (red) before and 500 s after stimulation. Black lines show standard deviation across neurons for a single simulation run. (D) Firing rate of all excitatory neurons in the network before (*top*) and 500 s after (*bottom*) stimulation. Neurons are displayed in a 32 x 50 grid. The black square on each panel indicates neurons belonging to the assembly. (E-F) Same as (C-D) for the CV. (C-F) Mean firing rate and CV are calculated using 50 s of activity. (G) Raster plots showing 3 s activity of 160 neurons within the assembly and 160 excitatory neurons outside of the assembly before stimulation (*top*) and 500 s after stimulation (*bottom*). The purple lines indicate neurons belonging to the assembly. (H) Firing rate of the readout neuron connected to the assembly (purple edge) or outside assembly (red edge), before and after stimulation. Black lines show standard deviation across 5 independent simulation runs.

The increase in excitatory-to-excitatory weights also led to higher correlation between assembly neurons (Fig 3G and S6 Fig). This means that the formed assemblies are 'silent' in terms of mean firing rate, but not in terms of correlation. The strength of within-assembly connectivity was determined here by the maximum weight allowed for excitatory-to-excitatory connections $W_{E \to E}^{max}$. If weaker assemblies were formed, the effect on CV was smaller than the observed one. If stronger assemblies were formed, on the other hand, there was an increase in correlation (S5 and S6 Figs). Here, we chose a fixed $W_{E \to E}^{max}$ that was large enough to promote a response in terms of change in CV, but that did not lead to strong correlations within the excitatory population (S6 Fig). In a metaplasticity framework, however, $W_{E \to E}^{max}$ could also be made plastic on a slower time scale than the iSTDP [2] and the triplet rule [13], which would allow for the formation of assemblies of different strengths.

As previously seen in the network without excitatory plasticity (S3(H) Fig), we also observed that the output neuron connected to the EI assembly fired with a larger firing rate than the output neuron connected to the random group of excitatory neurons (Fig 3H). This is

mostly due to higher correlation within assembly neurons, but higher CV could also contribute to this effect (S7 Fig).

In summary, in neurons belonging to an EI assembly, assembly embedding encodes a trace in the regularity of their spike trains, as well as on pairwise correlations within assembly neurons, which can be decoded by an output neuron through plastic connections with STF. Put together, this suggests that there are traces of the memory available from the neuronal activity even during seemingly silent moments, which could be potentially used for downstream processing without the need for an externally stimulated recall.

## Stronger assemblies decay more slowly due to both the irregular spiking and correlations

Due to the homeostatic nature of iSTDP, assembly neurons fired at target rate after formation of the assembly (Fig 3C and 3D). Given that this value was below the threshold for potentiation of the triplet rule, the stronger weights between assembly neurons decayed with time (Fig 4, S8 and S9 Figs). We were therefore interested in how the decay of excitatory weights was influenced by the strength of the assemblies. In order to test that, we performed the following simulations. After forming an assembly by stimulating a subgroup of neurons, the external input was set back to its baseline value and we measured the weight decay between pairs of synaptically connected neurons belonging to the assembly (Fig 4A). We performed separate simulations in which the assemblies were formed with different strengths, by setting different values of maximum allowed excitatory-to-excitatory weights $W_{E \to E}^{\max}$.

We observed that the stronger assemblies decayed more slowly than the weaker ones (Fig 4B and 4D). Slower decay of stronger assemblies could be explained by increased correlation

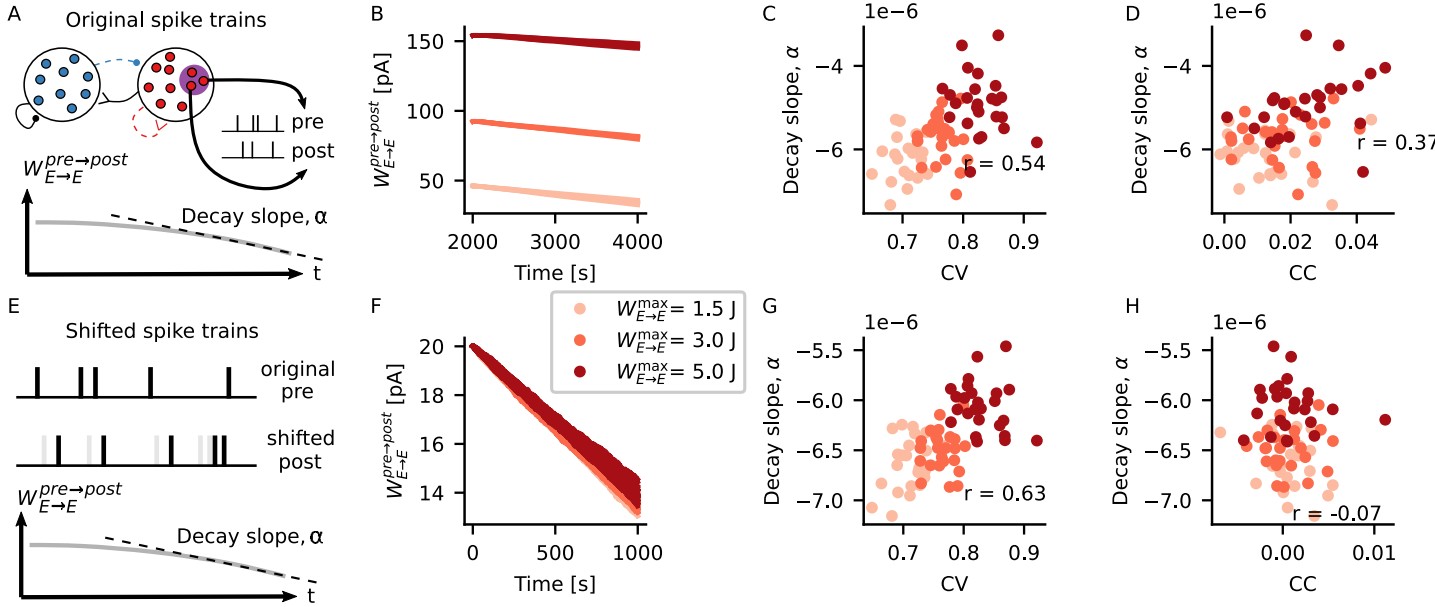

**Fig 4. Stronger assemblies decay more slowly due to both the irregular spiking and correlations.** (A) Spike trains of 5 pairs of synaptically connected assembly neurons and the time series of the excitatory weight between them are extracted from each of the 5 independent simulation runs shown on Fig 3. A linear function is fitted to the last 1000 s of the weight decay, and its decay slope $\alpha$ is extracted. (B) Excitatory weight between assembly neurons during 2000 s after the embedding of the assembly. Shown are the weights between 5 different pairs of connected neurons, for each of 3 different assembly strengths (shades of red, $W_{E \to E}^{\max} = 1.5$, 3 and 5 J). (C) Slope of weight decay $\alpha$ plotted against mean CV between pre- and post-synaptic spike trains for different strengths of assembly. (D) Slope of weight decay $\alpha$ plotted against correlation coefficient (CC) between pre- and post-synaptic spike trains for different strengths of assembly. (C-D) Mean CV and CC are calculated using the last 50 s of simulation, $r$ shows the Pearson's correlation coefficient between $x$ and $y$ values. (E) For each pair of pre- and post-synaptic neurons in (A), the post-synaptic spike train is shifted by 3 s, and the synaptic weight between the last 1000 s of the pre- and manipulated post-synaptic spike trains is calculated. (F-H) Same as (B-D), but for the shifted spike trains.

caused by the stronger excitatory weights (Fig 4D). Increased correlation means there should be a higher occurrence of pre-post pairs within short time windows. Considering the triplet rule [13], it is expected that such an increase should lead to more potentiation between excitatory weights and, therefore, slower decay of the assemblies. At the same time, the triplet rule is also known to potentiate post-pre-post triplets [13]. In that case, we should also expect more potentiation in cases where the postsynaptic neuron is firing with higher CV, given that higher CV translates into an increased occurrence of shorter intervals between two consecutive postsynaptic spikes.

Therefore, we also expected a slower decay of assembly weights due to higher CVs, and not exclusively due to increased correlation coefficient (CC). In order to test this, we tried to disentangle the effects of CC and CV by calculating the weight decay of manipulated spike trains. For each pair of pre-post spike trains recorded in our intact simulation (Fig 4A), we artificially shifted the post-synaptic spike train by 3 s, and calculated the weight decay for the manipulated spike train (Fig 4E). By performing this shift we observed two things. Firstly, as expected, shifting the spike trains led to lower correlation between spike trains (Fig 4H) and overall faster decay of assembly weights (Fig 4F–4H). Secondly, the weights of assemblies with higher CV decayed more slowly even for the shifted spike trains (Fig 4G), when correlation between pre and post activity was almost zero (Fig 4H). Taken together, these results indicate that not only CC, but also higher CVs contribute to slower decay of assemblies.

## Discussion

Cell assemblies are considered to be the substrate of memories in the brain. But how exactly memories are encoded in the synaptic weights and the firing statistics of assembly neurons is yet to be fully understood. Our results suggest that, in the presence of inhibitory plasticity [2], a memory trace can be encoded in the regularity of neuronal firing. More specifically, we have shown that increasing excitatory input to a neuron, which also received plastic feedforward inhibition [2], caused the neuron to fire spike trains with higher CV. We have also shown that this change in CV could be read out with the support of short-term facilitation [19]. In a recurrent network model with excitatory [13] and inhibitory [2] plasticity, we have shown that embedding a cell assembly left a trace in terms of irregular firing, which suggests the memory could still be available for influencing downstream processing even when the memory is stored in a silent state. Furthermore, we have shown that excitatory weights within the assembly decayed more slowly for stronger assemblies, due to both increased irregularity and increased correlation between assembly neurons.

Assembly formation is driven in our model by an increase in external input to assembly neurons. This increase in input leads to increased firing rate that triggers changes in excitatory and inhibitory synaptic weights. It is not completely clear, however, what exactly drives the formation of assemblies in the brain. It has been recently proposed that correlated input could drive assembly formation even in the absence of rate-based plasticity mechanisms [6] and that assemblies could also form spontaneously in the absence of structured external stimulation [7]. Future work could extend our current analysis and form silent EI assemblies without the transient increase in mean firing rate by stimulation of assembly neurons.

In our current work, within-assembly excitatory weights decayed back to baseline levels if the formed assembly was not reactivated by external input (Fig 4 and S8 Fig). This means that any memory encoded in the assembly would be slowly forgotten if no external reactivation was performed. A memory that is not forgotten without external reactivation would require that assembly weights are strengthened during spontaneous activity. Previous theoretical work has shown that in a network with multiple plasticity mechanisms, the structure of cell

assemblies can be reinforced during spontaneous activity through self reactivation [3]. Even without self reactivation, when the firing rate of assembly neurons is below the potentiation threshold for the triplet rule, within-assembly weights can still be reinforced during spontaneous activity if correlation between assembly neurons is high enough, as previously shown in [6]. In our simulations, stronger correlation between assembly neurons could be achieved with stronger excitatory weights, but this would possibly also require stronger recurrent weights overall in the network to stabilize network activity. Alternatively, including other plasticity mechanisms such as plastic excitatory-to-inhibitory weights could lead to the formation of balanced clusters containing both excitatory and inhibitory neurons. Embedding such clusters in networks has been shown to allow multistability and faster transitions in assembly activity between high and low firing rate [26]. In any case, the formation of self reinforcing silent EI assemblies through increased correlation between assembly neurons would also make assemblies less silent.

Silent assemblies have also been shown to form in simulations without inhibitory plasticity [27]. For example, silent assemblies can be formed with a model of structural plasticity on excitatory synapses [28, 29]. In those studies, stimulation of an assembly led to rewiring of synapses such that the assembly neurons were more likely to be connected, but the total indegree of excitatory neurons remained unchanged. Different to what we found here, the silent assemblies in that case were formed by rearranging the excitatory connections, but without increasing total excitatory and inhibitory input currents. Therefore, the mean excitatory and inhibitory input to assembly neurons was the same before and after the assembly embedding. In that case, embedding the assembly does not leave a trace on the regularity of firing of assembly neurons. Therefore, silent assemblies formed in different ways could potentially leave specific markers on neuronal firing patterns, which could contribute different functional aspects to a more complex circuit.

One prediction from our simulations is that neurons belonging to an engram fire with more irregular spike trains. While not many studies have investigated firing patterns of engram neurons *in vivo*, Tanaka et al. [30] did find that engram neurons were more likely to fire in bursts. They measured activity of place cells in hippocampal region CA1 during context discrimination. They found that not all place cells belonged to an engram, but those that did had higher burst rates and shorter inter-burst intervals. Furthermore, they found that engram neurons were more likely to fire during, and be phase locked to, fast gamma events. Since fast gamma oscillations correlate with inputs from entorhinal cortex [31], they have suggested that engram neurons in CA1 may be more responsive to inputs coming from this region. It remains an open and interesting question whether there could be any causal relationship between the burst firing of engram cells and their responsiveness to specific inputs.

Burst firing could also have an influence on how input signals are processed and on how signals are propagated to downstream areas [32]. In our model, the memory encoded in a silent EI assembly would be reflected in the CV, or level of burstiness of individual neurons. Interestingly, in a recent study, Koren et al. [33] found that the activity of bursty neurons in monkey primary visual cortex was more informative for decoding behavior than the activity of non-bursty neurons. Bursts have also been proposed to modulate the effect of plasticity [13, 14], and shown to implement credit assignment in a model of burst-dependent synaptic plasticity [34]. Burst coding could also be relevant to how neurons process both bottom-up and top-down information in hierarchical architectures. Naud and Sprekeler [25] have proposed that this problem could be solved by multiplexing using single spikes and bursts as two distinct codes. Similarly in the work we present in this paper, the firing rate and regularity of spike trains, or the CV, could also serve as two separate streams of information. Different to Naud and Sprekeler [25], however, the CV is modulated by the amount of inhibitory current, which

changes according to the iSTDP rule [2]. Therefore, changes in CV happen at a slow time scale. In other words, the signal encoded by the CV would have to be a slow signal. Alternatively, a faster signal could be constructed by gating the activity of different neurons, or populations of neurons, that fire with constant CV. On the other hand, faster changes in firing rate could still be propagated without triggering plastic changes through iSTDP and, therefore, without affecting the CV. In such a simplified model, however, the system would be limited to transmitting two streams of information, and transmitting more streams simultaneously would require a more complicated architecture. It remains an open question of how the synaptic increase and the larger CV within an EI assembly modulate the output firing of a stimulated neuron.

We have shown that a change in CV could be read out by a neuron connected to the assembly through STF (Figs 2 and 3). We did not study how the connections from the assembly to the readout neuron are formed. Here, by hardwiring those connections, we implicitly assumed those had been previously learned. Different memories could be stored in different assemblies that connect to different readout neurons. Memory recall could be implemented by direct stimulation of part of the assembly [2] or by disinhibition [17]. The STP, in this context, could allow for a modulatory signal that scales with the strength of within-assembly synapses and is available to downstream areas to be used even when the assemblies are silent, for example to stir plasticity [34]. Future work could explore how to combine short- and long-term synaptic plasticity in order to form these connections to the readout neurons.

In conclusion, our results show that embedding cell assemblies in a network with excitatory and inhibitory plasticity can leave a trace in terms of regularity of firing. This means that information about the assembly can be present in the neuronal activity even when the memory is stored in a silent state. Moreover, we showed that this information could be read out during spontaneous activity with support of STF, which suggests the silent memory could potentially modulate other signals in the absence of direct stimulation. Furthermore, we also showed how this change in regularity contributes to the longevity of memories. Put together, our results propose a different way in which memories could be encoded in silent EI assemblies.

# Methods

## Neuron model

All neurons in our simulations were current-based leaky integrate-and-fire (LIF) with exponential post-synaptic currents (PSC). The sub-threshold membrane potential $V_i$ of neuron $i$ obeyed the following equation:

$$\tau_m \frac{dV_i}{dt} = -V_i + RI(t),\tag{1}$$

where $\tau_m = 20$ ms is the membrane time constant and $R = 80$ MOhm is the input resistance. The input current $I(t)$ consisted of the sum of all excitatory and inhibitory currents coming from pre-synaptic sources. Unless stated otherwise, the input current from a pre-synaptic neuron $j$ to a post-synaptic neuron $i$ evolved according to:

$$\frac{dI_j(t)}{dt} = -\frac{I_j(t)}{\tau_{\text{syn}}} + W_{ij}\sum_k \delta(t - t_j^k),\tag{2}$$

where $\tau_{\text{syn}} = 1.5$ ms is the synaptic time constant and $W_{ij}$ represents the synaptic weight between pre-synaptic neuron $j$ and post-synaptic neuron $i$. The spike train $\sum_k \delta(t - t_j^k)$

consisted of all spikes produced by neuron $j$. The synaptic weight $W_{ij}$ was fixed for static synapses. For synapses that obeyed excitatory and inhibitory plasticity, $W_{ij} = \bar{w} \times w_{ij}(t)$, where $\bar{w}$ is a scaling constant and $w_{ij}(t)$ is a dimensionless variable that evolved according to the equations for excitatory and inhibitory plasticity described below. $\bar{w}_E = 1\,\text{pA}$ for excitatory synapses and $\bar{w}_I = -1\,\text{pA}$ for inhibitory synapses. In the following sections, some synaptic weight parameters are given with respect to a reference value $J = 30.8\,\text{pA}$, which was chosen such that the maximum amplitude of the post-synaptic potential would be 0.15 mV.

Every time the membrane potential reached a threshold value $V_{\text{th}} = 20\,\text{mV}$, the neuron emitted a spike. Following a spike, the membrane potential was reset to $V_{\text{reset}} = 10\,\text{mV}$ and remained there for a refractory period $t_{\text{ref}} = 2\,\text{ms}$. These parameter values were taken from [23].

## Plasticity models

**Inhibitory plasticity.**  Plastic inhibitory-to-excitatory connections followed the inhibitory spike timing-dependent plasticity rule (iSTDP) by [2]. In this rule, synaptic weights $w_{ij}$ between pre-synaptic neuron $j$ and post-synaptic neuron $i$ are updated whenever there is a pre-synaptic spike ($t^{\text{pre}}$) or post-synaptic spike ($t^{\text{post}}$), respectively, according to:

$$\begin{aligned} w_{ij}(t) &\rightarrow w_{ij}(t) + \eta(x_i - \alpha) \quad \text{if} \quad t = t^{\text{pre}}, \\ w_{ij}(t) &\rightarrow w_{ij}(t) + \eta x_j \quad \text{if} \quad t = t^{\text{post}}, \end{aligned} \tag{3}$$

where $\eta$ is the learning rate, $\alpha = 2 \times \rho \times \tau_{\text{STDP}}$ is a depression factor and $\rho$ is a constant parameter that sets the target firing rate of the post-synatic neuron [2]. The synaptic trace $x_i$ increases by 1 whenever neuron $i$ fires a spike and decays otherwise with time constant $\tau_{\text{STDP}}$, according to:

$$\frac{dx_i(t)}{dt} = -\frac{x_i(t)}{\tau_{\text{STDP}}}. \tag{4}$$

The parameters used were $\eta = 0.3$, $\rho = 9\,\text{Hz}$ and $\tau_{\text{STDP}} = 20\,\text{ms}$. Weights were bound to a maximum $W_{I \to E}^{\max} = 3\,000\,\text{pA}$.

**Excitatory plasticity.**  Recurrent excitatory-to-excitatory connections in the network simulations were plastic according to the triplet-based model of spike timing-dependent plasticity by [13]. In this model, synaptic weights $w_{ij}$ between pre-synaptic neuron $j$ and post-synaptic neuron $i$ are updated whenever there is a pre-synaptic spike ($t^{\text{pre}}$) or post-synaptic spike ($t^{\text{post}}$), respectively, according to:

$$\begin{aligned} w_{ij}(t) &\rightarrow w_{ij}(t) - o_1(t - \epsilon)[A_2^- + A_3^- r_2(t - \epsilon)] \quad \text{if} \quad t = t^{\text{pre}}, \\ w_{ij}(t) &\rightarrow w_{ij}(t) + r_1(t - \epsilon)[A_2^+ + A_3^+ o_2(t - \epsilon)] \quad \text{if} \quad t = t^{\text{post}}, \end{aligned} \tag{5}$$

where $A_2^-$, $A_3^-$, $A_2^+$, $A_3^+$ denote amplitude of weight changes and $r_1$, $r_2$, $o_1$ and $o_2$ are synaptic traces. In the original model [13], $\epsilon$ is a small positive constant to ensure weights are updated before the traces $o_2$ and $r_2$. In our simulations, $\epsilon$ illustrates the fact that weights were always updated before all trace values, including $o_1$ and $r_1$. The pre-synaptic (post-synaptic) traces $r_1$ and $r_2$ ($o_1$ and $o_2$) are increased by 1 whenever the pre-synaptic (post-synaptic) neuron fires,

and decay otherwise according to:

$$\begin{aligned}
\frac{dr_1(t)}{dt} &= -\frac{r_1(t)}{\tau_+}, \\
\frac{dr_2(t)}{dt} &= -\frac{r_2(t)}{\tau_x}, \\
\frac{do_1(t)}{dt} &= -\frac{o_1(t)}{\tau_-}, \\
\frac{do_2(t)}{dt} &= -\frac{o_2(t)}{\tau_y}
\end{aligned} \tag{6}$$

Weights were bounded between $W_{E \to E}^{\min} = J$ and $W_{E \to E}^{\max}$, which was assigned different values at different simulations. The parameters used were taken from [13]: $A_2^- = 7 \times 10^{-3}$, $A_3^- = 2.3 \times 10^{-4}$, $A_2^+ = 7.5 \times 10^{-10}$, $A_3^+ = 9.3 \times 10^{-3}$, $\tau_+ = 16.8$ ms, $\tau_x = 101$ ms, $\tau_- = 33.7$ ms, $\tau_y = 125$ ms.

**Short-term plasticity.** In simulations with short-term plasticity, the model used was the short-term facilitation (STF) by [19]. In this model, the total synaptic input to a post-synaptic neuron $i$ is given by:

$$I(t) = \sum_j A y_j(t), \tag{7}$$

where $A$ is the absolute synaptic weight, and $y_j$ determines the effective contribution of the PSC from neuron $j$ to the input current to neuron $i$. It evolves according to the system of equations:

$$\begin{aligned}
\frac{dx_j}{dt} &= \frac{z_j}{\tau_{\text{rec}}} - u_j x_j \delta(t - t_{pre}), \\
\frac{dy_j}{dt} &= -\frac{y_j}{\tau_{\text{syn}}} + u_j x_j \delta(t - t_{pre}), \\
\frac{dz_j}{dt} &= \frac{y_j}{\tau_{\text{syn}}} - \frac{z_j}{\tau_{\text{rec}}},
\end{aligned} \tag{8}$$

where $x_j$, $y_j$ and $z_j$ are the fraction of synaptic resources in the recovered, active and inactive states, respectively, from neuron $j$, $t_{\text{pre}}$ denotes the timing of a pre-synaptic spike, $\tau_{\text{syn}}$ is the decay time constant of PSC and $\tau_{\text{rec}}$ is the recovery time constant for depression. The variable $u_j$ describes the effective use of synaptic resources by each pre-synaptic spike, and it evolves according to:

$$\frac{du_j}{dt} = -\frac{u_j}{\tau_{\text{fac}}} + U(1 - u_j)\delta(t - t_{\text{pre}}) \tag{9}$$

where $\tau_{\text{fac}}$ is the time constant for facilitation and the parameter $U$ determines how much $u_j$ is increased with each spike. The absolute synaptic weight used was $A = 1000$ pA. The remaining parameters were taken from [25]: $U = 0.02$, $\tau_{\text{rec}} = 100$ ms, $\tau_{\text{fac}} = 100$ ms.

## Simulations

All simulations were performed using the neural network simulator NEST 2.20.0 [35].

### Single neuron simulation (Fig 1)

**Spiking simulation.** A single output neuron received input from an external input, an excitatory and an inhibitory source. The external input represented a source of feedforward input and it was modeled as a Poisson process with rate $v_{\text{ext}} = 18$ kHz. It connected to the output neuron with a fixed synaptic weight $W_{\text{ext}} = J/3$, which did not change throughout the simulation. The excitatory source represented recurrent input received from other neurons within the same network. It was modeled as a Poisson process with rate $v_{\text{exc}} = 1440$ Hz and it connected to the output neuron with a fixed synaptic weight $W_{E_1 \rightarrow E_2}$, which was set to $W_{E_1 \rightarrow E_2} = J$ during a warm-up period, and increased afterwards. The inhibitory source was modeled as a Poisson process with rate $v_{\text{inh}} = 360$ Hz, and it connected to the output neuron with a plastic synapse following the iSTDP rule. The choices of parameters were made in order to match the scenario from Fig 3. After a warm-up period of 200 s of simulation, the excitatory synaptic weight was increased to a multiple of the original weight $W_{E_1 \rightarrow E_2} = 1J, 2J, 3J, 4J, 5J$. The weight from the external input source remained unaltered. The simulation ran for another 200 s. Mean firing rate and coefficient of variation of inter-spike intervals of the output neuron were calculated using the last 50 s of simulation.

**Subthreshold membrane potential simulation.** The mean and variance of the subthreshold membrane potential ($x$ and $y$ coordinates of red crosses in Fig 1E and 1F and $x$ coordinates on Fig 1G) were calculated in a new set of simulations. In those simulations, the spiking threshold of the output neuron was removed, such that the output neuron produced no spikes. Given that the output neuron produced no spikes, the connection from the inhibitory source was static. The weight $W_{I \rightarrow E}$ used was the mean synaptic weight from the spiking simulation, averaged across the last 100 s of simulation. This scenario was simulated for 400 s, and mean and standard deviation of membrane potential were calculated using the last 200 s of simulation.

### Single readout simulation (Fig 2)

Two output neurons received input from 160 input sources. Both neurons were the same, except that the spiking neuron had a spiking threshold $V_{\text{th}} = 20$ mV and the non spiking neuron had none. Input sources were modeled as Gamma processes. Their spike trains were generated by randomly sampling inter-spike-intervals from a Gamma distribution with parameters shape $k = \frac{1}{\text{CV}^2}$ and scale $\theta = \frac{\text{CV}^2}{v}$, where CV was the prescribed coefficient of variation of inter-spike intervals and $v = 9$ Hz was the prescribed mean rate of each spike train. A different value of CV was used for each simulation, ranging from CV = 0.4 to CV = 1.4 in intervals of 0.1. Both output neurons also received a constant current $I = 150$ pA. Each simulation lasted 55 s. Output rate was calculated from the spiking output neuron using the last 50 s of simulation. The mean and standard deviation of subthreshold membrane potential was calculated from the non spiking output neuron using the last 50 s of simulation.

In the simulations with short-term plasticity, the input sources connected to the output neurons with plastic synapses following STF. In the simulations with no plasticity, the input sources were connected to the output neurons with a fixed synaptic weight that was equal to the mean synaptic weight of the STF simulations with CV = 0.4.

### Network simulation (Figs 3 and 4)

The recurrent network comprised $N_E = 1600$ excitatory and $N_I = 400$ inhibitory neurons. The excitatory (inhibitory) population formed synapses to randomly selected neurons from both excitatory and inhibitory populations with an indegree $C_E = 0.1N_E$ ($C_I = 0.1N_I$). All neurons

received a background input in the form of a spike train with Poisson statistics with rate $v_{ext}$ = 18 kHz and weight $W_{ext} = J/3$. Synapses from the excitatory to the inhibitory population were static with weight $W_{E \rightarrow I} = J$. Synapses from the inhibitory to the inhibitory population were static and stronger by a factor of 10 ($W_{I \rightarrow I} = -10J$). Excitatory-to-excitatory synapses followed the triplet based STDP rule, and inhibitory-to-excitatory synapses followed the iSTDP rule.

After a warm-up period of 2000 s, a subgroup comprising 10% of the excitatory neurons was stimulated. Stimulation consisted of increasing the rate of the external input to the stimulated subgroup by a factor 5 for 1 s. Following stimulation, the rate of the external input was set back to its original value $v_{ext}$, and the network was simulated for further 2000 s.

Two readout neurons received input from either the stimulated neurons, or a subgroup of excitatory neurons with the same size as the stimulated subgroup. Both readout neurons also received a constant current $I$ = 150 pA. The synapses connecting excitatory neurons to readout neurons followed STF.

The network simulations were run 5 times with different random seeds.

## Theoretical rate and CV

The firing rate $v$ of a leaky integrate-and-fire neuron can be estimated by the following equation (see details of the derivation in [23, 24]).

$$v = \left[ t_{ref} + \tau_m \sqrt{\pi} \int_{\frac{V_{reset} - \mu}{\sigma}}^{\frac{V_{th} - \mu}{\sigma}} e^{u^2} (1 + \mathrm{erf}(u)) du \right]^{-1} \quad (10)$$

where $\mu$ and $\sigma$ are respectively the mean and standard deviation of the subthreshold membrane potential, $t_{ref}$ is the refractory period, $\tau_m$ is the membrane time constant of the neuron, $V_{th}$ is the threshold potential, $V_{reset}$ is the reset potential and erf() is the error function.

The coefficient of variation of inter-spike intervals for a neuron firing with rate $v$ and different combinations of mean $\mu$ and variance $\sigma$ of subthreshold membrane potential can be theoretically predicted using the following equation (see derivation in [23]):

$$\mathrm{CV} = \left[ 2\pi v^2 \int_{\frac{V_{reset} - \mu}{\sigma}}^{\frac{V_{th} - \mu}{\sigma}} e^{x^2} dx \int_{-\infty}^{x} e^{y^2} (1 + \mathrm{erf}(y))^2 dy \right]^{\frac{1}{2}} \quad (11)$$

## Data analysis

**Firing rate.** Mean firing rates $r$ were calculated using:

$$r = \frac{S}{N \Delta T}, \quad (12)$$

where $S$ is the number of spikes of all $N$ neurons during time interval $\Delta T$. For single neuron mean rate, $N$ = 1. $\Delta T$ = 50 s unless stated otherwise.

**CV.** Coefficient of variation of inter-spike intervals (ISI) were calculated using:

$$CV = \sigma_{ISI} / \mu_{ISI}, \quad (13)$$

where $\sigma_{ISI}$ is the standard deviation and $\mu_{ISI}$ is the mean of the ISI of an individual neuron. CVs were calculated using 50 s of spiking data.

**CC.** The spike count correlation between a pair of neurons $i$ and $j$ was calculated as the Pearson correlation coefficient

$$R_{ij} = \frac{c_{ij}}{\sqrt{c_{ii}c_{jj}}}, \qquad (14)$$

where $c_{ij}$ is the covariance between spike counts extracted from spike trains of neurons $i$ and $j$, and $c_{ii}$ is the variance of spike counts extracted from neuron $i$. In Fig 4, correlations were calculated from spike trains comprising the last 1000 s of activity, using bins of size 10 ms. In S6 Fig, correlations were calculated from spike trains comprising the 10 s of activity shown in the raster plots, using bins of size 10 ms.

**Decay slope.** The decay slope $\alpha$ of excitatory weights was calculated by fitting a linear function to the last 1000 s of the $W_{E \rightarrow E}(t)$ decay data, and extracting its slope. The fitting was performed using a standard fitting algorithm from NumPy.

## Supporting information

**S1 Fig. Effect of static synaptic weights on the read out firing rate as a function of CV.** (A) Distribution of synaptic weights from the simulations in Fig 2E–2H when CV = 0.4 (left) and CV = 1.4 (right). (B-G) Same as Fig 2, but static synaptic weights were chosen to be equal to the mean of the plastic weights when CV = 1.4.
(EPS)

**S2 Fig. Effect of increasing CV on the firing rate of a readout neuron receiving short-term facilitating inputs.** A readout neuron received input from 160 neurons through plastic connections with short-term facilitation. Each input neuron was modeled as a spike train generated as follows: the first 50 inter-spike intervals (ISI) were drawn from a Gamma distribution with parameters $k = \frac{1}{CV^2}$ and scale $\theta = \frac{CV^2}{\nu}$, where $\nu = 9$ Hz and $CV = 0.7$, and the last 50 inter-spike intervals were drawn from a Gamma process with same rate $\nu = 9$ Hz but a CV = 0.8. The readout neuron also received a constant current $I = 150$ pA. (A) Average firing rate of the readout neuron calculated with bins of 100 ms. (B) Average synaptic weight across all 160 inputs calculated with bins of 100 ms. (A-B) Shown in the mean weight and firing rate across 500 independent simulation runs. (C) Raster plot of the 160 inputs. Grey (red) dots are the spikes drawn from the distribution corresponding to CV = 0.7 (CV = 0.8). The red vertical line indicates the time of the first spike from the distribution with CV = 0.8 (red).
(EPS)

**S3 Fig. Formation of assembly without the triplet rule.** Same as Fig 3 from the main text, but excitatory to excitatory connections are static (no triplet rule). The assembly is formed by hardwiring an increase in excitatory weights between assembly neurons by a factor of 6. Please note that in order to achieve a similar effect on CV, the increase by a factor of 6 is larger than $W_{E \rightarrow E}^{max} = 5$ J on the main figure. This is because the triplet rule leads to potentiation of weights from all excitatory neurons in the network to the assembly neurons, which is not the case for this static scenario (Compare (H) to Fig 3H in the main text).
(EPS)

**S4 Fig. Time series of weights, rates and CV for the assembly simulation (Fig 3).** (A) Mean excitatory-to-excitatory weights between assembly neurons. (B) Mean inhibitory weights onto assembly neurons. (A-B) Mean weights were calculated across all weights and across 5 independent simulation runs. (C) Mean firing rate of assembly neurons. (D) Mean CV of assembly neurons. (C-D) Mean firing rate and CV were calculated across all assembly neurons and

across 5 independent simulation runs. (E) Mean firing rate of the readout neuron connected to the assembly, calculated across 5 independent simulation runs. (A-E) The grey shaded area indicates the moment when the stimulus was on. All weights and firing rates were calculated using bins of 100 ms and CV was calculated with bins of 1 s. Please note that estimating CV from such small time bins could lead to under estimation of its value due to under sampling of large inter-spike-intervals.
(EPS)

**S5 Fig. Formation of stronger assembly.** Same as Fig 3 from the main text, but with $W_{E \to E}^{max} = 5.5$ J.
(EPS)

**S6 Fig. Whole population raster plot.** (A) Raster plot of all assembly neurons (purple) and all other excitatory neurons outside the assembly, during 10 s before (*top*), and 500 s after (*bottom*) stimulation, for $W_{E \to E}^{max} = 5$ J (same simulation as Fig 3 in the main text). (B) Same as (A) for $W_{E \to E}^{max} = 5.5$ J (same simulation as S5 Fig). (C) Average cross-correlation between assembly neurons (within, purple), between non-assembly neurons (outside, red) and between one assembly and one non-assembly neuron (between, grey). Shown is the mean Pearson's correlation coefficient at different time lags, averaged across all pairs.
(EPS)

**S7 Fig. Effect of correlation on the firing rate of the readout neuron.** The spike trains from all within and outside assembly neurons connected to readout neurons were extracted from the simulations in Fig 3. Each of the spike trains was shifted by a random lag, drawn from a uniform distribution between 0 s and 3 s. The shifted spike trains were then used as input to a readout neuron which was the same as in Fig 3. (A) Firing rate of the readout neuron receiving the original (assembly) and shifted (assembly shifted) spike trains from the assembly neurons or from 160 excitatory neurons outside the assembly (outside and outside shifted). Shifting the spike trains led to no response of the readout neuron to either within or outside assembly neurons, indicating that higher correlations did contribute to higher readout response. In order to test whether there is any effect from the increased CV of the assembly neurons on the readout firing rate, we increased the constant current the readout neuron received by 20%. (B) Same as in (A), but the readout neuron received a constant current of 180 pA (instead of 150 pA). In that case, shifting the spike trains led to a decrease in the readout response to both within and outside assembly neurons. However, the readout response to the assembly neurons was still higher than the response to neurons outside the assembly, indicating that higher CV led to stronger readout response. Bars indicate mean and standard deviation across 5 independent simulation runs.
(EPS)

**S8 Fig. Decay of assembly weights to baseline.** Excitatory-to-excitatory weights between assembly neurons as a function of time for different values of $W_{E \to E}^{max}$. For these simulations, plasticity was accelerated by multiplying $\eta$ from the iSTDP rule, and $A_2^+, A_2^-, A_3^+, A_3^-$ from the triplet rule by a factor of 10. Shown are the weights between 5 different pairs of pre- and post-synaptic neurons, for each simulation run.
(EPS)

**S9 Fig. Time series of CV.** (A) CV of neurons within the assembly (within, purple) and outside the assembly (outside, red) during the simulations in Fig 3 as a function of time, for $W_{E \to E}^{max} = 1.5$ J (left), 3 J (middle) and 5 J (right). CV was estimated for each neuron using bins of 1 s. Shown in the average across neurons, across 5 independent simulation runs. Please note

that estimating CV from such small time bins could lead to under estimation of its value due to under sampling of large inter-spike-intervals. (B) Same as in (A) for the simulations with accelerated plasticity (S8 Fig).
(EPS)

## Acknowledgments

We thank Douglas Feitosa Tome for his comments on the manuscript and members of the Clopath Lab for insightful discussions.

## Author Contributions

**Conceptualization:** Júlia V. Gallinaro, Claudia Clopath.

**Data curation:** Júlia V. Gallinaro.

**Formal analysis:** Júlia V. Gallinaro.

**Funding acquisition:** Claudia Clopath.

**Investigation:** Júlia V. Gallinaro, Claudia Clopath.

**Methodology:** Júlia V. Gallinaro, Claudia Clopath.

**Software:** Júlia V. Gallinaro.

**Supervision:** Claudia Clopath.

**Validation:** Júlia V. Gallinaro.

**Visualization:** Júlia V. Gallinaro.

**Writing – original draft:** Júlia V. Gallinaro.

**Writing – review & editing:** Claudia Clopath.

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
