## [Decision Letter · Decision Letter 0]

26 Aug 2021

Dear Dr. Gallinaro,

Thank you very much for submitting your manuscript "Memories in a network with excitatory and inhibitory plasticity are encoded in the spiking irregularity" for consideration at PLOS Computational Biology. As with all papers reviewed by the journal, your manuscript was reviewed by members of the editorial board and by several independent reviewers. The reviewers appreciated the attention to an important topic. Based on the reviews, we are likely to accept this manuscript for publication, providing that you modify the manuscript according to the review recommendations.

Sincerely,

Michele Migliore

Associate Editor

PLOS Computational Biology

Samuel Gershman

Deputy Editor

PLOS Computational Biology

[LINK]

Reviewer's Responses to Questions

**Comments to the Authors:**

Reviewer #1: This paper explores how memories can be encoded in the spiking irregularity of "silent" cell assemblies -- that is, cell assemblies with low mean firing rates. Quiescence in such assemblies is maintained through EI balance, and it has traditionally been thought that memories could be retrieved from these assemblies by momentarily disrupting this balance, either through excitation or disinhibition from an external input. This paper proposes the novel idea that silent assemblies can also encode for memories through the irregularity of their spiking pattern rather than just the firing rates alone. Indeed, this irregularity could allow downstream neurons to read out the memory without needing to disrupt the EI balance, making such a mechanism energetically more efficient.

The authors show that after an assembly is momentarily stimulated by external inputs, EI balance will help maintain the assembly's mean firing rate at the same low level as the background population (hence their "silence"), but the assembly neurons' spiking will be more irregular compared to their non-assembly counterparts (irregularity is quantified by the coefficient of variance or CV). This mechanism depends on both excitatory and inhibitory plasticity, both to encode the new memory and to maintain EI balance after encoding. In single neuron simulations, they show that different inputs into a single neuron will lead to different levels of irregularity but the same mean firing rate, suggesting that spiking irregularity can serve as a mechanism for coding different inputs. They also show how short-term plasticity offers a mechanism for decoding the level of irregularity in assemblies. And they finish by exploring how spiking irregularity can contribute to the longevity of memories.

Overall, this paper is clearly written and easy to follow. The main ideas are well-supported by a series of simulations with LIF neurons that follow a logical progression. And while this reviewer is not an expert on the memory literature, I did find both the context of this work and how their contribution fits in with previous work on the subject well explained. Indeed, I think the clarity is a strength and would help this paper reach a wider audience. I do have a few suggestions for how this paper could be further strengthened:

* in the simulation for Fig 2, the authors compared a simulation with fixed synaptic weights against one with plastic weights. The fixed weights (W_{E->E}) were set to J. I am wondering what range of values (e.g. the mean and variance) did the plastic weights cover. Was this significantly different from J? If so, it may be helpful if the authors also ran a simulation where the fixed weights were better matched at least to the mean value of the variable weights.

* in the network simulations, the authors only gave one measurement for the assembly CV and one measurement for the readout firing rate. To read out the spiking irregularity of a neuron, a downstream neuron would presumably need to integrate input spikes across a time window. To get a sense of how quickly a downstream neuron can reliably measure the irregularity of an upstream neuron, it would be helpful to plot time courses of the CVs as well as of the readout firing rates, preferably on the same time axis.

* similarly, in Fig 4, the authors only plotted the time course of the weights. Since memory is manifested as spiking irregularity, it would also be informative to plot the time course of the CV -- this would provide a more direct measurement of a memory's effective decay rate, since downstream neurons have access only to this information and not to assemblies' synaptic strengths.

* in line 140, the authors commented on how assembly neurons were more correlated and pointed to the raster plot of Fig 3G as evidence. Rather than just relying qualitatively on raster plots, I think it'd be better if the authors could provide a more quantitative measure of correlation here. One option is the Pearson correlation that the authors used in Fig 4, but I think temporal cross-correlations can also be informative, as this would give a better sense of how correlated spike timings are. For this, the authors can consider three types of cross-correlations: average cross-correlations between assembly neurons, average cross-correlations between an assembly neuron and a non-assembly neuron, and average cross-correlations between non-assembly neurons.

* in lines 149-150, the authors speculate that the larger firing rate of the assembly's readout neuron is due to both the assembly's higher CV as well as higher correlations. The speculation about the correlations is reasonable, and there is a simple test the authors could perform to help validate this. The authors can introduce a random lag to each assembly neuron's spike train to break up the correlations, and then feed these lagged spike trains into the readout neuron. A lower firing rate would support the hypothesis that correlations do contribute to the higher firing rate.

* in the Discussion, it could be helpful if the authors would comment on the fidelity of memory retrieval with their set-up. While the authors did show in a simple example how different coefficients of variability can lead to different firing rates in readout neurons, it is not immediately clear how this would scale to the encoding and decoding of more complex inputs. Specifically, two questions come to mind. Can different inputs get encoded with similar patterns such that an STP-based readout would have difficulty distinguishing one memory from the other? And as the synpatic weights decay, would the original memory start resembling other memories? While I appreciate that a thorough investigation of these questions is a subject for future work, it'd be nice to have some preliminary discussion of it in the paper.

Minor points:

* line 75: presumably CV means the coefficient of variation, but the acronym should be defined explicitly when first used

* Fig 1G: the 1Hz and 8 Hz lines have very similar colours making it difficult to tell which graph is which. I presume the straighter graph is the 1Hz one.

* line 141: “raster” rather than “rater”

* line 324 and 327: it is said that the external synaptic weight W_ext = J/3 did not change across simulations while the synaptic weight W_E->E = J did vary. This is confusing as it sounds like W_ext = W_E->E / 3 which both changes and does not change. While we can probably guess what the authors actually mean, the authors should clarify.

Reviewer #2: Please find my review attached.

**Have the authors made all data and (if applicable) computational code underlying the findings in their manuscript fully available?**

Reviewer #1: Yes

Reviewer #2: Yes

PLOS authors have the option to publish the peer review history of their article (what does this mean?). If published, this will include your full peer review and any attached files.

Reviewer #1: No

Reviewer #2: No

Figure Files:

Data Requirements:

Reproducibility:

References:

---

## [Editor Report · Decision Letter 1]

26 Oct 2021

Dear Dr. Gallinaro,

We are pleased to inform you that your manuscript 'Memories in a network with excitatory and inhibitory plasticity are encoded in the spiking irregularity' has been provisionally accepted for publication in PLOS Computational Biology.

Best regards,

Michele Migliore

Associate Editor

PLOS Computational Biology

Samuel Gershman

Deputy Editor

PLOS Computational Biology

---

## [Editor Report · Acceptance letter]

5 Nov 2021

PCOMPBIOL-D-21-01370R1 

Memories in a network with excitatory and inhibitory plasticity are encoded in the spiking irregularity

Dear Dr Gallinaro,

I am pleased to inform you that your manuscript has been formally accepted for publication in PLOS Computational Biology. Your manuscript is now with our production department and you will be notified of the publication date in due course.

With kind regards,

Katalin Szabo
